# Agent Productivity Modeling in a Call Center Domain Using Attentive Convolutional Neural Networks

**DOI:** 10.3390/s20195489

**Published:** 2020-09-25

**Authors:** Abdelrahman Ahmed, Sergio Toral, Khaled Shaalan, Yaser Hifny

**Affiliations:** 1Department of Electronics Engineering, University of Seville, 41092 Seville, Spain; storal@us.es; 2Faculty of Informatics, The British University in Dubai, Dubai 345015, UAE; khaled.shaalan@buid.ac.ae; 3Faculty of Computer Sciences and Information, Helwan University, Helwan 11795, Egypt; yhifny@fci.helwan.edu.eg

**Keywords:** productivity modeling, LSTMs, CNNs, attention layer

## Abstract

Measuring the productivity of an agent in a call center domain is a challenging task. Subjective measures are commonly used for evaluation in the current systems. In this paper, we propose an objective framework for modeling agent productivity for real estate call centers based on speech signal processing. The problem is formulated as a binary classification task using deep learning methods. We explore several designs for the classifier based on convolutional neural networks (CNNs), long-short-term memory networks (LSTMs), and an attention layer. The corpus consists of seven hours collected and annotated from three different call centers. The result shows that the speech-based approach can lead to significant improvements (1.57% absolute improvements) over a robust text baseline system.

## 1. Introduction

Productivity evaluation of an agent in call centers is a challenging task. Human subjective evaluations are dominating approaches to the current systems [1,2]. The subjective factors drastically bias and deviate the evaluations from the real or actual performance when they depend on an agent’s tone, communications skills, oral proficiency, and listening skills [3,4,5]. The subjective evaluation is called finesse standards that allow for style and individuality and provide room for interpretation [6]. The objective performance (productive) refers to individual proficiency and relevant activities that contribute to the organizational ’technical core’ [7]. The technical core, in call centers, refers to standards that the agent follows supported by relevant technical knowledge in responding to customer inquiries. The technical core standards like following the call scripts and predefined scenarios when they start with a greeting, verifying the caller account, and the responding to the customer inquiries correctly.

The second challenge is that the evaluation process is crucial when it depends on manual assortment, considering the evaluations are massive over a certain period, i.e., one year. Call centers are dynamic and rushed with customer contacts over various communication channels [6]. Automating the subjectivity detection and productivity evaluation is essential in reducing effort and time for the manual evaluation process. Furthermore, the manual evaluation is subjected to a high probability of errors, which reflects on the overall performance of the organization. Recently, objective methods have been developed to overcome the limitations of subjective evaluations [8,9]. These methods are based on a text-based approach. The calls were transcribed into text using a speech recognition system. Hence, they are classified into productive/nonproductive according to a binary classification model [10]. The evaluation of the productivity of an agent from the speech signal directly is significantly different from the text-based approach. The extracted features from the recorded call are massive when compared to the text features [11]. Besides, the text-based approach requires a robust speech recognition engine that may limit the accuracy of the classifier. On the other hand, the drawback of speech signal processing is that it requires more computational resources than the text classification approach.

The contribution of the paper is to propose an evaluation framework based on different classification approaches using speech signal processing for accuracy improvement. The acoustic features extracted from a recorded call may carry much information when compared to its transcribed text. For this reason, we employ sophisticated models for classification based on convolutional neural networks (CNNs) and long-short-term memory networks (LSTMs) [12,13]. Our goal is to classify the speech audio segments of an agent into productive and nonproductive (i.e., binary classification problem). The research tries to prove whether the speech signals carry more informative features than antecedent text classification models.

The structure of the paper is organized as follows: Section 2 details the previous related works about productivity in call centers. The proposed framework for productivity modeling is described in Section 3. The training data description is elaborated in Section 4. The experiment setup is detailed in Section 5. The study results are discussed in Section 6 and finally Section 7 concludes the paper results.

## 2. Related Work

Previous related studies on productivity modeling were based on text classifications [14,15]. Ahmed et al. [9] transcribed the calls into text, manually and by using a speech recognition engine [16], and then classified them according to a pre-annotated corpus with a binary classifier (productive/nonproductive). The productivity modeling was based on the generative model, Naive Bayes, by determining the posterior probability of a productivity class conditioned on observations [9]. A similar study was conducted based on a discriminative approach [8]. The Logistic Regression and Linear Support Vector Machine (LSVM) were used to improve the classification accuracy. There were other applications using speech processing for detecting eminent factors of the customer complaints and classifying them for the better conclusion of the relevant causal roots [17]. Big data analytics application has been applied to the recorded calls for detecting the quality of service delivered to the customer. It was based on Hadoop Map Reduce framework and utilize text similarity algorithms such as Cosine and n-gram [18]. It also integrated slang word lists to the monitoring system. Perera et al. [19] developed a software for the automatic handling of the call center agent performance. They propose predefined factors like speech rate, voice intensity level, and emotional state in order to evaluate the performance of contact center agents, and applied a Support Vector Machine (SVM). The classification is limited to these predefined factors, which requires more investigation into other hidden factors. Another Software developed to evaluate the call center representative based on machine learning and language processing [20]. It used several transcription systems APIs (google, wit, sphinx) in order to analyze the performance based on emotions, banned words, greeting words, and the usage of competitors’ names.

The productivity measurement based on speech processing is inspired by emotion recognition studies [21,22,23]. However, productivity is different from the emotional behavior of the agent. For instance, the agent of the call center has to follow the call script, which starts with a greeting, introducing him/herself, and finally, the list of common questions and answers that regularly comes to the call center. By focusing on the speech segments of the agent, we propose to classify them into productive and nonproductive categories for determining the link with the series of actions after excluding the customer part. Combining both CNNs and LSTMs gives an outstanding improvement in emotional and speech recognition studies [22,24]. The convolutional layers extract salient features, and the long-short-term memory (LSTMs) layers handle the sequential phenomena of the speech signal. The layers are followed by an attention layer, which extracts a summary vector that is fed into a binary classifier. The attention weights, generated within the attention layer, explore the call segment frames that contribute to determining the most informative part in each segment [25]. The next sections demonstrate the way of combining different modeling approaches in order to improve the modeling accuracy compared to the antecedent text classification approaches.

## 3. The Proposed Framework

The proposed framework is based on four different classification models: (1) CNNs, (2) CNNs-Attention and (3) CNNs-LSTMs, and (4) CNNs-LSTMs-Attention. The objective of the framework is to find the best setup in terms of accuracy and performance. Figure 1 illustrates the framework model.

The agent’s speech segments are extracted based on a diarization process. A diarization process aims to split the speech based on the speakers of a call into different voice segments [26]. The diarization is required to eliminate the customer segments in order to evaluate only the agent’s segments, which are the concerned subject. The next step is to convert the voice utterances/signals into Mel-Frequency Cepstral Coefficients (MFCCs) format [27,28]. The third step is the model training for the four approaches mentioned. The output accuracy will be compared to determine the delta between the models.

### 3.1. CNNs and BiLSTMs

CNNs have been proven to be significant in speech recognition and many other signal processing models [29]. The CNNs help to squash the frequencies’ redundancy through the filters. Hence, they extract salient features through efficient computational algorithm in parallel mechanism. The LSTM is a form of recurrent neural networks (RNNs), which handle the gradient vanishing problem [30]. In the case of RNNs, the gradient becomes very small for long sequences, which prevents the weights from being updated.

The LSTMs can find longer temporal dependencies than simple RNNs. However, the LSTMs layers are slow in processing the input sequence of speech frames. A variant of LSTMs known as bidirectional LSTMs (BiLSTM) [31] allows the integration of both past and future information for better accuracy than legacy LSTMs. It is a combination of two LSTMs in two directions: One operates in the forward direction, and the other operates in the backward direction. Hence, each input frame at time *t* is aware of the past and future contexts, which improves its accuracy. The output of the CNNs or BiLSTMs layers are passed to a global max-pooling layer or an attention layer to convert the sequence of frames into a summary vector. Different models’ combinations will be applied in this study to compare their classification performance.

The 1D-CNNs architecture consists of four layers with 500 filters, five kernel sizes each, ReLU activation function. The CNNs-BiLSTMs is 256 filters for layer-1, 64 filters for layer-2, two layers of BiLSTMs with 128 units each. A Max-pooling layer is required to downsample the CNNs output and reduce the dimensions to learn. The dense layer is 64 units to convert the dimensions of the vectors and forward them to the output layer. The output layer is a logit sigmoid for binary classification (productive/nonproductive).

### 3.2. Attention Layer

The CNNs or the BiLSTMs layers generate a sequence of vectors for the classification process [32]. The attention layer is used to convert the sequence of vectors (frames) into a context vector, which attends some parts of the input sequence [33,34]. Figure 2 illustrates the role of the attention layer in our approach.

The modeling layers are indicated in solid gray.The attention layer is the dotted box, including the circles that represent the calculation of the attention weights. The Softmax function calculates the attention weights and generates the context vector *C*.The context vector is fed into a dense layer with a tanh activation function.The output layer is the logit regression (sigmoid) function for the segment classification.

The attention weight, using Softmax function, is the probability of frame contribution at time *t* to the remaining frames in the same segment. For each frame vector $x_{t}$ in a sequence of inputs $x_{1},x_{2},\ldots ,x_{T}$, the attention weights $\alpha_{t}$ are given by:(1)$$\alpha_{t}=\frac{\exp(f\left(x_{t}\right))}{\sum_{j=1}^{T}\exp\left(f\left(x_{j}\right)\right)}$$
where $f(x_{t})$ is defined using the trainable parameters w as follows:(2)$$f\left(x_{t}\right)=\tanh\left(w^{T}x_{t}\right)$$

The context vector *C* is the summation of the attention weights multiplied by the input frames.
(3)$$C=\sum_{t=1}^{T}\alpha_{t}x_{t}$$

The Dense layer output *D* is tanh is given as follows:(4)$$D=tanh(W^{T}C+b)$$
where *W* are the weights of the hidden layers, and *b* is the bias. The output layer for classification is the Logit function over two classes (productive/nonproductive).
(5)y=Logit(D)

## 4. The Data

The experimental study is performed on the corpus of a real estate call center’s speech data. The corpus consists of seven hours over 30 calls (14 min per call on average) for six different agents (40% females and 60% males). It is considered adequate compared to another similar study [22]. It was collected over the phone with a sampling rate of 8 kHz. The real estate call center is intended for sales and marketing activities. The agent talking time is around 85% of the call (almost 6 h) after removing the customer part using the diarization process. Call center experts in customer service performed the annotation through a focus group. Krippendorff’s Alpha has been applied to measure the degree of agreement among multiple raters when manually annotating the call segments by a human to avoid evaluation bias. The agreement for the annotation (productive/nonproductive) should be around 80% (Alpha > 0.8) among the raters in order to consider the classification valid. The raters’ agreement in the current study is almost achieved (79%) [35].

## 5. The Experiment

The study experiment is performed on the corpus of a real estate call center’s speech data. The Data is diarized using the SIDEKIT tool (s4d for short) [26]. The diarization is performed by cascaded methods to get the optimum results: Bayesian Information Criterion (BIC.), Hierarchical Agglomerative Clustering (HAC), and then Viterbi decoding, respectively. The segmentation process sometimes fails to detect the speakers because of the low sampling rate (8 kHz). Hence, the diarization has been performed on two steps: The first step is upgrading the sampling rates to 16 kHz for extracting the segmentation boundaries. The second step is applying the segments information extracted from the previous step on the 8 kHz corpus. The training was performed on balanced voice segments of the agent for measuring the performance after excluding the customer segments.

Essentia toolkit [36] has been used to extract 13 MFCCs due to a low sampling rate and less complexity to generated models. The audio signal is segmented into 25 ms frames with a 10 ms shift. Five cross-validations are applied to verify model accuracy. Each approach, mentioned in Table 1, generates five accuracies where the average of the five accuracies presents the concluded model accuracy. The training is performed over two NVidia GPUs Quadro M4000 and M5000. The classification models were built using Keras 2.2.5 and TensorFlow 1.4 based on the Conda environment. Table 1 summarizes the parameters of experiments’ setups as per Figure 1.

## 6. The Results

The models have been trained and validated with five cross-validations with the accuracy percentage stated in Table 2. The experiment leads to the following analysis themes:

### 6.1. CNNs-BiLSTMs vs. CNNs Accuracy

The resulted accuracy of the BiLSTMs layer is higher than the CNNs model as expected. It proves that the BiLSTMs can handle a more extended sequence of data for processing when compared to CNNs. However, the processing time of CNNs is one-fourth of the processing time of BiLSTMs on the same setup. BiLSTMs consume much more time because of its recurrent units. The accuracy improvement using BiLSTMs when compared to CNNs is less than 1% (0.85%), which makes it questionable if a longer time and more resources are worth it for a minor improvement.

### 6.2. The Attention Layer Effect

Referring to Table 2, it is clear the effect of the attention layer on CNNs model with an improvement of (1.57%). Nevertheless, in the case of CNNs-BiLSTMs, there is no effect of the attention layer as the results before and after adding the attention layer are almost the same. Moreover, the CNNs-attention accuracy is higher than the CNNs-BiLSTMs-attention approach, which is unexpected. The delta is (0.73%), which is due to the presence of the attention layer that concludes the improvement of the attention layer in CNNs setup compared to BiLSTMs.

### 6.3. The Attention Layer and Most Informative Frames

The attention layer focuses on the significant frame out of the frames sequence for classification. The attention weights per time are illustrated in Figure 3.

The figure describes the attention weights curve for a sample segment of the call. The *x* axis is frames per call, and the *y* axis represents the attention weights as an output of the Softmax function. The high weight means the classifier pays attention to a significant frame out of the remaining sequence. By analyzing the peaks for all segments’ graphs, a better knowledge of those features that impact productivity during the call can be obtained. The first observation is irrelevant features to productivity like the drop call cadence, customer talk, and the crowd noise. Customer talk means a small portion that was not excluded accurately through the diarization process. The crowd noise is the background noise of other agents in the call center’s workplace. The graph, in Figure 3, illustrates two peaks at frames 70 and 155 of the cross talk of the customer and the agent speech. The second observation is that there are significant features such para−linguistic features in the call. The features are summarized in the following points:Stuttering "Umm Ahh": This is the common attention for nonproductive calls, which is repeated during the call with an average duration from one to two seconds.The tone level: The high tone triggers the high attention for productive calls. The proper tone level is an important factor in call centers that indicates the wakefulness and enthusiasm of the agent. The primary reason for a frustrated customer is the insincere tone of voice from the person handling the query [37].

### 6.4. The Speech vs. Text Classification

The baseline of previous studies related to text classification and accuracy is stated in the Table 3. The speech models achieve a significant improvement of 1.57% compared to the highest accuracy of the text classification method using Linear Support Vector Machine (LSVM). For speech classification approaches, the models’ accuracies are very close to each other because of the limited calls that were processed (seven hours). Accordingly, we expect the accuracy gap increases for a more extended corpus. When it comes to modeling performance based on processing time, text modeling is much faster than speech. The text training takes less than five minutes compared to three hours for CNNs and 24 h in CNNs-BiLSTMs (speech approaches). However, it is essential to mention that speech recognition is mandatorily required before text classification. The automatic call transcription is a comprehensive process that consumes around 30 h for transcription extraction [16] (The experiments have been performed on a very close setup in performance to the current study) depending on the approaches used (HMM/GMM (Gaussian mixture model), RNN.). Keeping the same hardware setup, the speech model is higher performance than others when it consumes less time for training. The CNNs (45 min) is considered a baseline performance of 100% (least time). The remaining models’ performance is calculated by determining the ratio of CNNs processing time to the model time. Table 4 indicates the less time means higher performance for each model. For example, the CNNs-BiLSTMs-ATT model performance equals 0.75 h divided by 25.5 h, which results in a performance of 2.9%. The time and corresponding performance are illustrated for each text and speech approaches in Figure 4.

## 7. Conclusions

Productivity modeling is challenging due to the subjectivity of performance evaluation. The study proposed a productivity classification model based on speech signal processing for plenty of features and a simple setup. The experiment has been performed over seven hours of real estate call center speech. The accuracy achieved is 84.27%, which shows around a 1.57% improvement over text classification methods. The CNNs with an attention layer provides faster and better accuracy than CNNs-BiLSTMs models. The delta improvement between CNNs and CNNs-BiLSTMs models are less than 1%. The study recommends using the CNNs-Attention layer instead of CNNs-BiLSTMs and CNNs-BiLSTMs-Attention models because of the small differences in accuracy and less processing time. The recommended future work is combining the speech and text classification models for better classification results compared to this study. The multi-classes model may extend the benefits of this study based on the availability of a precisely annotated dataset. Using prosodic feature extraction instead of MFCC may give rich information and better classification than MFCC vocal tract features. Furthermore, the attention layer informs about the para-linguistic productivity factors that may carry a significant contribution to the call center domain.

## Figures and Tables

**Figure 1 sensors-20-05489-f001:**
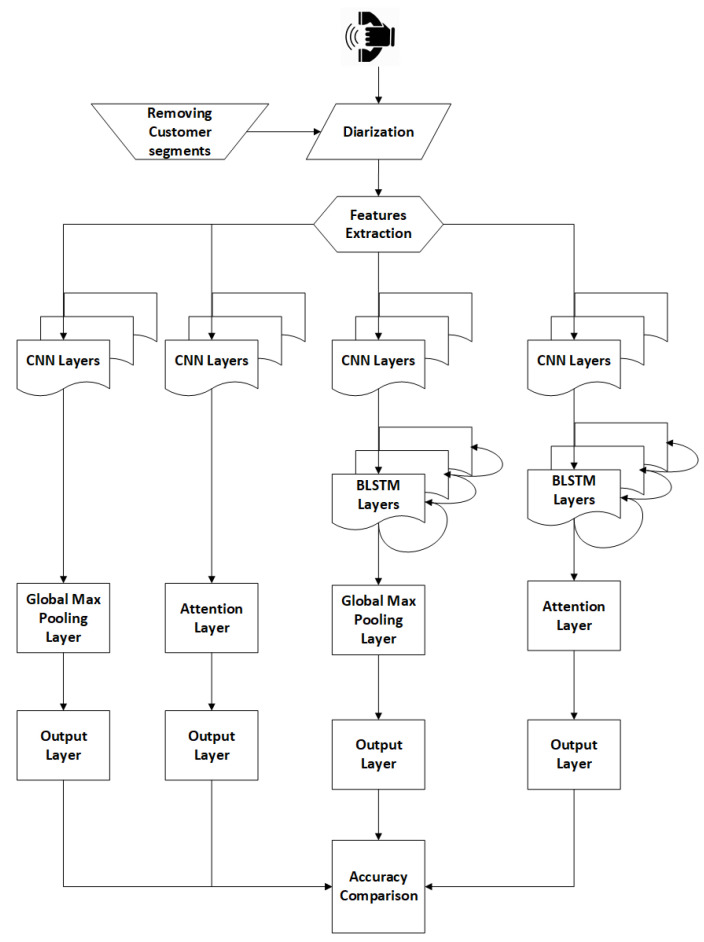
The proposed Framework.

**Figure 2 sensors-20-05489-f002:**
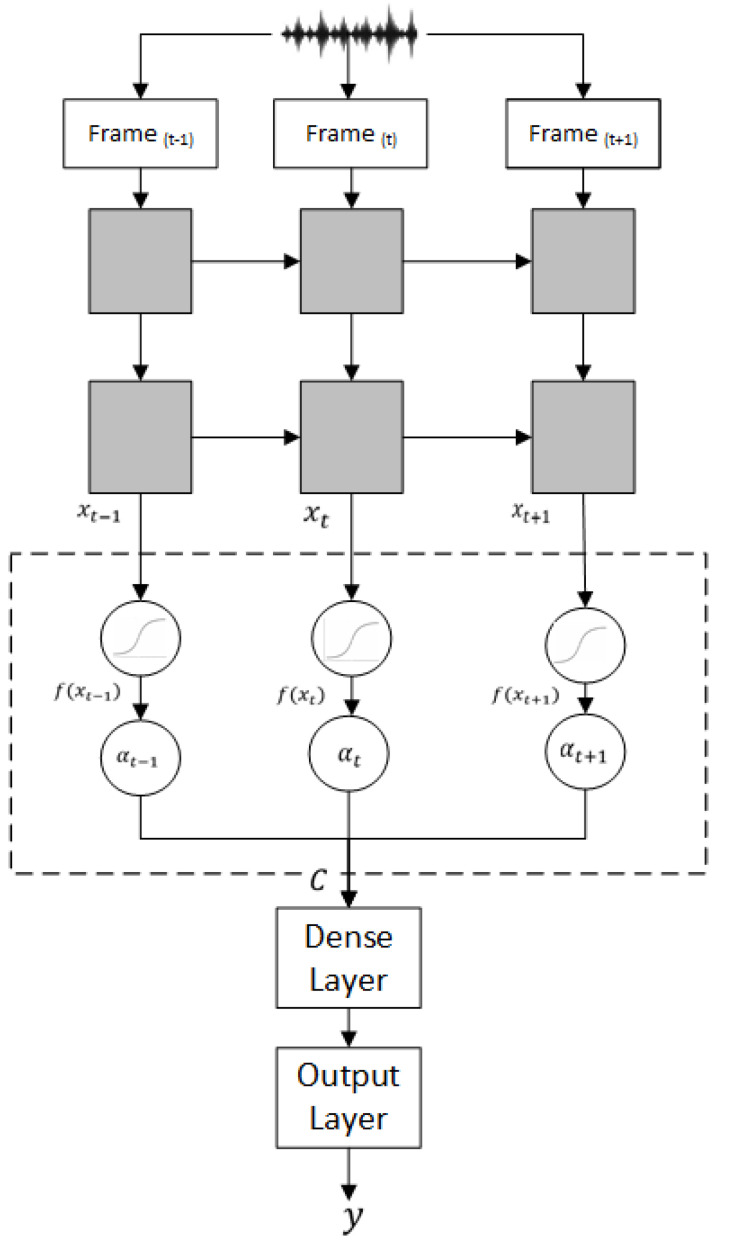
The role of the attention layer.

**Figure 3 sensors-20-05489-f003:**
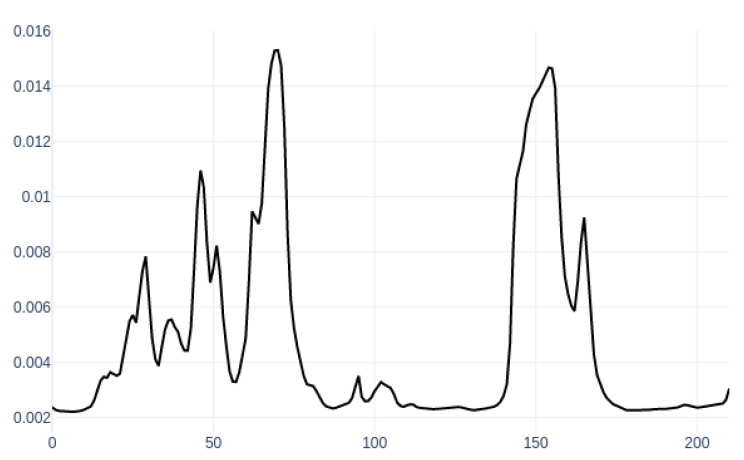
An attention graph (frames vs weights) for a segment of speech.

**Figure 4 sensors-20-05489-f004:**
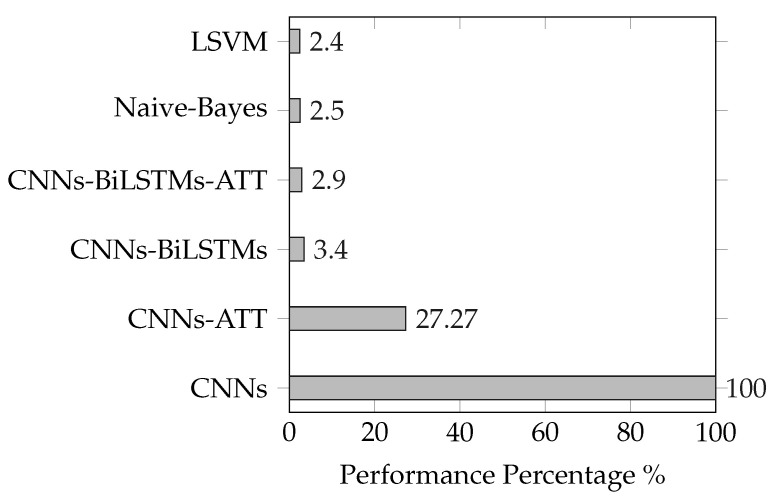
Y axis:Models/Models indices, X axis is the Relative performance for the models based on CNNs performance baseline/Modeling Time.

**Table 1 sensors-20-05489-t001:** Convolutional neural network (CNN) vs CNN-long-short-term memory network (CNN-LSTM) Filters/Units.

Model Type (Filters-Units)				
Layer	CNNs	CNNs-Att	CNNs-BiLSTMs	CNNs-BiLSTMs-Att
Input	13	13	13	13
CNNs-1	500	500	256	256
Max Pooling-1	-	-	256	256
CNNs-2	500	500	64	64
Max Pooling-2	-	-	64	64
CNNs-3	500	500	-	-
CNNs-4	500	500	-	-
BiLSTMs-1	-	-	128	128
BiLSTMs-2	-	-	128	128
Attention	-	500	-	128
Global Max Pooling	500	-	128	-
Dense	500	500	64	64
Output (Classifier)	2	2	2	2

**Table 2 sensors-20-05489-t002:** Different models Accuracy.

Accuracy % per Model Type				
Fold	CNNs-BiLSTMs	CNNs	CNNs-Att	CNNs-BiLSTMs Att
1	81.97%	78.4%	82.5%	80.2%
2	81.39%	83.7%	83.72%	80.2%
3	84.3%	83.7%	83.72%	84.2%
4	84.8%	81.9%	83.72%	87.2%
5	85.3%	85.96%	87.71%	85.9%
Average	83.55%	82.7%	84.27 %	83.54%

**Table 3 sensors-20-05489-t003:** Accuracy (Speech–Text) comparison.

Accuracy % per Model Type		
Classification Method	Type	Accuracy
Naive Bayes	Text	67.3%
Logistic Regression	Text	80.76%
Linear Support Vector Machine (LSVM)	Text	82.69%
CNNs	Speech	82.7%
CNNs-BiLSTMs-Attention	Speech	83.54%
CNNs-BiLSTMs	Speech	83.55%
CNNs-Attention	Speech	84.27%

**Table 4 sensors-20-05489-t004:** Performance-Time Based comparison.

Performance % per Model Type		
Classification Method	Processing Time (Hour)	Performance %
LSVM	31	2.4%
Naive Bayes	30	2.5%
CNNs-BiLSTMs-Attention	25.5	2.94%
CNNs-BiLSTMs	22	3.4%
CNNs-Attention	2.75	27.2%
CNNs	0.75	100%

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
