# Peer review of "Agent Productivity Modeling in a Call Center Domain Using Attentive Convolutional Neural Networks"

_sensors, 2020, doi:10.3390/s20195489_

Round 1

Reviewer 1 Report

The paper explores several designs for measuring the productivity of an agent in a call center. These methods are convolutional neural networks (CNNs), long-short-term memory networks (LSTMs), and an attention layer. The authors used a corpus consists of seven hours over 30 calls (14 minutes per call on average) for six different agents (40% females and 60% males). According to the obtained results, the authors state that the accuracy achieved is 84.27%, which shows around a 1.57% improvement over text classification methods, but a comparison with other methods is not shown in the paper (maybe the comparison is located in the missing tables).

Authors should replace the "affiliation" word in the paper's header with their respective universities.

On line 107, the C variable must be mentioned in the Text. ("The context vector C").

There is no information about Author Contributions, Funding, Acknowledgments, and Conflicts of Interest. This information should be placed at the end of the paper, after the references section.

Reviewer 2 Report

The authors propose a framework for modeling agent productivity for real estate call centers based on speech signal processing. The problem is formulated as a binary classification task and the authors explore several designs for the classifier.
The topic is very interesting and also the comparison between the different implemented approaches.

However, there are some problems in this paper. I list the problems as follows.

1) The manuscript lacks clarity at some places.
  - Section 2 should be expanded with other more recents works in this field.
  - Subsection 3.1 is too generic. It shoud be contains a detailed description of the network. For example, the sentence inserted in the Section 5, lines 133-138, could be moved in this section.
More details about the CNN architecture must be provided, also with a table that reports the layers type and description.
  - In Subsection 3.1 the authors mentioned the BiLSTMs, but it's not clear if this variant of LSTM has been used in the framework. If not, why introduce this sentence in the description of the framework? Please clarify this point.
2) The English needs improvement.

Minor comments:
- In the Introduction, line 37, is missing the description of Section 4.
- The first and second author belong to the same affiliation.
- There are several typos to fix, specially in the References.

Round 2

Reviewer 1 Report

All comments have been attended

Reviewer 2 Report

The authors responses are adequate to the comments/suggestions raised.
There are no further comments and suggestions.